# Tunable Optimal Dual Band Metamaterial Absorber for High Sensitivity THz Refractive Index Sensing

**DOI:** 10.3390/nano12152693

**Published:** 2022-08-05

**Authors:** Madurakavi Karthikeyan, Pradeep Jayabala, Sitharthan Ramachandran, Shanmuga Sundar Dhanabalan, Thamizharasan Sivanesan, Manimaran Ponnusamy

**Affiliations:** 1Department of Communication, School of Electronics Engineering, Vellore Institute of Technology, Vellore 632014, India; 2Department of Electronics and Communication Engineering, Sri Manakula Vinayagar Engineering College, Puducherry 605107, India; 3School of Electrical Engineering, Vellore Institute of Technology, Vellore 632014, India; 4Functional Materials and Microsystems Research Group, Royal Melbourne Institute of Technology University, Melbourne, VIC 3001, Australia; 5School of Computer Science and Engineering, Vellore Institute of Technology, Vellore 632014, India; 6School of Electronics Engineering, Vellore Institute of Technology, Chennai 600127, India

**Keywords:** absorbers, biosensors, glucose sensors, high sensitive sensors, dual band absorbers

## Abstract

We present a simple dual band absorber design and investigate it in the terahertz (THz) region. The proposed absorber works in dual operating bands at 5.1 THz and 11.7 THz. By adjusting the graphene chemical potential, the proposed absorber has the controllability of the resonance frequency to have perfect absorption at various frequencies. The graphene surface plasmon resonance results in sharp and narrow resonance absorption peaks. For incident angles up to 8°, the structure possesses near-unity absorption. The proposed sensor absorber’s functionality is evaluated using sensing medium with various refractive indices. The proposed sensor is simulated for glucose detection and a maximum sensitivity of 4.72 THz/RIU is observed. It has a maximum figure of merit (FOM) and Quality factor (Q) value of 14 and 32.49, respectively. The proposed optimal absorber can be used to identify malaria virus and cancer cells in blood. Hence, the proposed plasmonic sensor is a serious contender for biomedical uses in the diagnosis of bacterial infections, cancer, malaria, and other diseases.

## 1. Introduction

The extraordinary qualities of graphene have inspired investigators to learn more about its many potential uses [1,2,3]. The most interesting property of graphene is the perfect absorption of terahertz (THz) radiation, which makes it suitable for applications such as sensors [4], detectors [5], cloaking devices [6], thermal emitters [7], modulators [8], and so on. The ease in tuning the graphene operating frequency by varying the electric field or chemical doping paves the way for designing the narrow and multi-band absorbers suitable for high sensitivity THz sensing. Recently, researchers have been looking into the use of ultrathin [9], ultrasensitive absorption-based sensors [10,11,12], and biosensors with a narrow [13] and ultra-narrowband response [14,15,16]. THz sensors have wide range of medical applications [17,18]. They are also called optical biosensors, which detect biomolecules directly in real-time [19]. Fluorescence and label-free detection are two techniques of sensing using the optical biosensors. The label-free technique, which uses surface plasmon resonance (SPR), is a low cost method compared to the fluorescence method [20]. The interaction of incident photons and free electrons on the surface of the metal results in the non-radiative electromagnetic mode is known as surface plasmon resonance (SPR). On the medium and metal surface, it is an excited state that spreads locally. It is only present on the surface of the medium or metal, where the surface intensity is greatest, and it gradually weakens on both sides in the direction perpendicular to the surface [21,22,23]. A wide-band absorber made up of a metal plate and a patterned vanadium dioxide film separated by a dielectric layer is proposed in [24], in which the temperature control enables the absorption intensity to be changed from 0 to 0.999. The absorber structure is the basis of these sensors. The absorbers having a sensitive absorption spectrum and narrow bandwidth are used to produce high sensitivity sensors [25].

Numerous graphene based absorbers have been proposed with multilayer structures and wideband absorption. One of the strong interactions that has recently been included into the metasurface system is bound states in the continuum (BIC), which is capable of encapsulating electromagnetic wave energy in the metasurface without radiation leakage [26]. To lessen plasmonic wave power loss, Vanani et al., have introduced a double waveguide THz sensor made of graphene and GaAs. Instead of using the standard double waveguide mode, they have placed a tiny ring in place of one of the waveguides to increase the structure’s sensitivity by using constrained supermodes [27]. In [28], a graphene based multi-layer absorber structure was proposed. By using a Al_2_O_3_ layer between two patterned graphene layers, four narrow absorption peaks are achieved. However, the bands have a large full width half maximum (FWHM) range, and the patterned graphene is more difficult to fabricate than the graphene sheet. In [29], using different patterns in three layers, a wideband absorber was proposed. It exhibits an absorption range of 0.3 THz to 3 THz. Using cross-over elliptic dielectric structures above the graphene layer, a wideband absorber was reported in [30]. However, the structure is not simple to fabricate. It was suggested in [31] that it is hard to fabricate the absorber with a multilayer structure. A narrowband absorption peak with simple structure was reported in [25]. The proposed structure exhibits only one narrowband. Hence, it is desired to have a simple structure absorber with narrow and multi-band absorption peaks for efficient sensing.

With this motive, some THz sensors have been proposed using graphene [32] and using other methods [33,34]. A gold ring and a graphene disk was used to produce Fano type resonances, thereby enabled ultrasensitive sensing in [35]. In [10], a high figure of merit (FOM) and sensitivity sensor was proposed by integrating a Fabry-Perot cavity and graphene based gratings.

In this study, a rather simplistic and compact absorber structure with a dual absorption peak for sensing applications is suggested. A single layer of graphene and an incredibly thin layer of SiO_2_ substrate with a thickness of 3 μm make up the suggested absorber. On top of the graphene sheet, there are four gold bars. Dual narrow band absorption peaks are produced by this configuration. Due to the narrow bandwidth of both of the dual absorption peaks, it is suitable for sensing applications. The structure of this paper is as follows: First, we describe the basic construction of the absorber and how different parameters affect the absorption peaks. Second, a highly sensitive sensor application based on the absorber has been suggested and evaluated. The dual narrow-band, uncomplicated structure, tunability, and controllability of the proposed absorber are its key benefits.

## 2. Design Structure and Model

In this section, the proposed absorber unit cell structure is depicted with a single sheet of graphene and four small sized gold bars. Figure 1a shows the 3D view of the proposed absorber. It consists of a gold metasurface array at the top. Gold bars with height $h_{g}$ and width w are placed in the form of a 2×2 array. Beneath the gold bars and above the substrate there is a graphene layer which has a thickness $t_{g}$ and chemical potential ($\mu_{c}$). Due to a dependence of the graphene surface plasmon’s (GSPs) properties on the substrate, the graphene layer is grown on a silicon dioxide SiO_2_ substrate with thickness $h_{s}$ and $\varepsilon_{r}=2.25$ [10]. Hence, the GSPs properties and absorption peaks will vary for a small variation in dielectric [36,37]. The GSPs structure’s key benefits are variation in its conductivity by bias voltage or chemical doping, low loss, and tunability [38,39]. To lessen the transmission, a gold layer with thickness $h_{g}$ and conductivity $\sigma =4.56\times 10^{7}S/m$ [31] is used.

Figure 1b shows the top view of the absorber. The distance between the gold bars in the x −axis and y −axis is kept at $d_{g}$ and $y_{g}$, respectively. At the centre, the graphene layer is placed with a dimension of $l_{g}\times w_{g}$. The increased absorption in SiO_2_ is due to the high conductivity of the graphene.

The Fabry-Perot cavity between gold layers and graphene traps the incident wave that passes through SiO_2_ layer. As shown in the Figure 1a, the incident wave is applied in the *z*-axis. To excite the GSPs, the incident wave’s electric field is polarised in the x-direction at the graphene-substrate interface. Due to GSPs and incident wave interaction, perfect absorption is achieved by graphene. Furthermore, the high confinement of GSPs makes the Fabry-Perot cavity resonance strong and results in the perfect absorption by the proposed absorber. The Kubo’s formula can be used to estimate the conductivity of the graphene as follows:(1)$$\sigma \left(\omega \right)=\frac{je^{2}\mu_{c}}{\pi h\left(\omega +j\tau^{-1}\right)}+\frac{je^{2}}{4\pi h}\ln\left(\frac{2\left|\mu_{c}\right|-h\left(\omega +j\tau^{-1}\right)}{2\left|\mu_{c}\right|+h\left(\omega +j\tau^{-1}\right)}\right)$$
where $e=1.6\times 10^{-19}\;C$, ω, $\mu_{c}$, τ, and h are the electron charge, angular frequency, graphene chemical potential, relaxation time, and reduced Planck constant. We assume 1 ps as the value of the relaxation time [40]. Graphene supports surface plasmons by acting as a thin metal layer due to the graphene’s positive imaginary part conductivity and dominant intra-band, i.e., the first term in (1) [10,41]. Using [40], the graphene tenability can be obtained as follows:(2)$$\mu_{c}=hv_{f}\sqrt{\frac{\pi \varepsilon_{0}\varepsilon_{r}V_{g}}{et_{g}}}$$
where $V_{g}$ is the external bias voltage and $v_{f}=\frac{10^{6}m}{s}$ is the Fermi velocity. From (2), it is obvious that by varying $V_{g}$, the $\mu_{c}$ can be varied. This allows for the graphene responding in many ways. Table 1 provides the optimum parameters for the required absorption peaks.

## 3. Results and Analysis

The proposed gold metasurface based absorber for sensor application is simulated using CST Microwave Studio software based on three dimensional (3D) FDTD techniques. The unit cell’s z-direction is surrounded by an open boundary condition, whereas, in the x and y direction, the periodic boundary condition has been used. The absorption coefficient (A) is determined using $A=1-{\left|S_{11}\right|}^{2}-{\left|S_{21}\right|}^{2}$, where $\left|S_{11}\right|$ and $\left|S_{21}\right|$ are the reflection coefficient and transmission coefficients, respectively. The evolution process is represented by the three stages in Figure 2. Figure 3 depicts an improvement in absorption peaks for different evolutionary phases from Figure 2. The figure shows that, in the absence of gold bars, the absorption weakens at f=11.7 THz. After placing the gold bars, nearly 100% absorption is found with two discrete resonant peaks. The first peak with a resonance frequency of 5.1 THz has an FWHM of 0.3223 THz, where FWHM is full width at half maximum. Here, FWHM corresponds to 50% absorption. As per the definition of quality factor $\left(Q\right)$ [4,42,43], the first peak Q value is 15.823. The resonance frequency of the second peak is observed at 11.7 THz, with an FWHM of 0.308 THz. The corresponding second peak Q value is 38.407. The second peak is narrower and has a high Q value when compared with the first peak. The simple structure absorbers in the literature do not possess two discrete narrowbands with high Q value.

Further, to evaluate the dependency of resonance frequency on structural parameter and to produce the optimized design, parametric analysis has been carried out. The proposed absorber will have a perfect absorption if the Fabry-Perot cavity resonance is coupled to GSP resonance. This can be achieved through the optimal selection of $\mu_{c}$. Hence, the effect of $\mu_{c}$ on the resonance is given in Figure 4. It can be noticed that at $\mu_{c}=0.5$, the absorption is maximum at two frequencies, i.e., f=5.1 and 11.7 THz. The other values of $\mu_{c}$ have resulted in imperfect absorption. However, it has been reported in [25] that, through proper selection of $w_{g}$ and $\mu_{c}$, any desired resonance frequency can be achieved.

As mentioned, $w_{g}$ impacts the electromagnetic wave confinement in the absorber. Hence, the effect of $w_{g}$ is studied and depicted in Figure 5. It is noticed that by varying $w_{g},$ the desired frequency of operation can be achieved. However, the second resonance, i.e., f=11.7 THz, shifts with the diminished absorption.

Hence, $w_{g}=5\;\mu m$ is chosen in the final design. Next, the effect of variation in substrate height $h_{s}$ is analysed and presented in Figure 6. On changing $h_{s}$, the absorption bandwidth is varied at the cost of reduced absorption. This can be observed from the Figure 6. The optimized value is chosen as $h_{s}=3,$ since it provides a compromise between narrow bandwidth and perfect absorption at two resonance frequencies.

The impact of incidence angle on the absorption spectrum is seen in Figure 7. Although the suggested absorber depends on the polarisation of the incoming wave, it is clear from the figure that it is independent of incident angles below 80°. Figure 8 depicts the sensing process using an analyte medium. The properties of graphene surface plasmons change with the changing refractive index of the analyte medium. Hence, we analysed different analyte thickness in THz region using the proposed absorber.

The frequency shift of the proposed sensor depends on the analyte thickness. It can be seen from Figure 9 that a maximum shift in frequency occurs when $t_{a}=2\;\mu m$. When the thickness is increased further, the shift in absorption becomes saturated. Hence, we have chosen $t_{a}=2\;\mu m$ for analysing the sensing capabilities of the proposed dual band absorber.

Figure 10 shows the shift in resonance frequency when the refractive index changes from n=1 to n=2. It can be observed that without much deterioration in absorption, there is a shift in absorption frequency. This proves that refractive index sensing can be performed with the help of the presented absorber structure.

Figure 11 displays the observed electric field magnitude ($\left|\vec{E}\right|$) distribution at the absorption frequencies of 5.1 THz and 11.1 THz. It seems to be that the electric component of the incident wave is orthogonal to the graphene and dielectric interface, where the GSPs are stimulated. As can be seen in Figure 5, the excited GSP and, consequently, the absorption structure are affected by the value of $w_{g}$.

## 4. Refractive Index Sensing of Glucose

Sensing quality of THz sensors can be measured by two main parameters, sensitivity (S) and figure of merit $\left(FOM\right)$. The sensitivity (S) can be found using $S=\Delta f/\Delta n\;\left(\mathrm{THz}/\mathrm{RIU}\right)$. The figure of merit is determined using FOM=S/FWHM. Here, *FWHM* is full width at half maximum. The refractive index of water is $n_{w}=1.3198$ and that of 25% glucose in water is $n_{gw}=1.3594$ [37]. For the analyte thickness of $t_{a}=2\;\mu m$, we have obtained the value mentioned in Table 2. It can be noticed that the proposed dual band absorber structure has attained a sensitivity of 2.08 THz/RIU and 4.72 THz/RIU in sensing water with 25% glucose at I and II resonant peak, respectively. This high value of sensitivity is due to the narrow bandwidth of the absorption spectrum.

## 5. Detection of Malaria

Around 250 million individuals worldwide contract malaria each year, making quick malaria screening essential [44]. A healthy red blood cell (RBC) has a refractive index of 1.399. Additionally, as malaria progresses, an infected RBC’s refractive indices are 1.383 and 1.373 in the trophozoite and schizont phases, respectively. As demonstrated in Table 3, the proposed sensor is capable of detecting various stages of malaria. The previous section claimed that a 2 μm analyte thickness offers good sensing capability. In order to identify malaria, we therefore set the analyte thickness to 2 μm. The proposed sensor has comparably achieved a high sensitivity value of 1.76 THz/RIU and 3.72 THz/RIU at the I and II resonant peak, respectively.

The proposed sensor can be fabricated using standard techniques. Additionally, because it has undergone extensive research, the recommended sensor has the capacity for ultra-high sensing for a variety of applications, including the detection of water contaminants or the detection of malaria.

In Table 4, a comparative analysis is shown. Recent reported structures in the literature are compared to the suggested work. Comparing the suggested absorber to recently reported alternative sensors, it performs better in terms of sensitivity, FOM, and Q-factor. In addition, the suggested tunable absorber offers a narrow geometry with superior angular and polarisation stability.

Recent fabrication and reporting of several biosensors based on metamaterials demonstrates a strong agreement between the results of simulation and measurement [42,51,52]. The belief that the absorber/sensor described in this article will function as anticipated if experimentally implemented in the future is reinforced by this latest research [42,51,52]. In addition, this structure can be constructed using a clear and precise method that is fully described in [53]. Using a shadow mask during the graphene’s plasma enhanced chemical vapour deposition (PECVD), the suggested method makes it possible to fabricate graphene in any shape on a SiO_2_ substrate. Other strategies and approaches for producing graphene with dimensions less than a nanometer have been researched and are presented in [54].

## 6. Conclusions

In this study, we suggest a THz absorber for sensor use. Due to the confinement of the graphene surface plasmon, the suggested absorber exhibits an extremely precise and condensed absorption peak. The simulation findings demonstrate the response’s tunability by adjusting the graphene chemical potential, which aids in having a perfect absorption at a specific frequency. Additionally, it is demonstrated that the response is insensitive to the angles of incidence up to 80°. With a high sensitivity of 4.72 THz/RIU, FOM of 13.88, and high Q value of 32.49, the proposed absorber can be a suitable candidate for bio-medical applications. This sensor is a suitable choice for THz sensing, such as glucose detection and malaria illness diagnosis, because of its simple design, ease of production, and good responsiveness.

## Figures and Tables

**Figure 1 nanomaterials-12-02693-f001:**
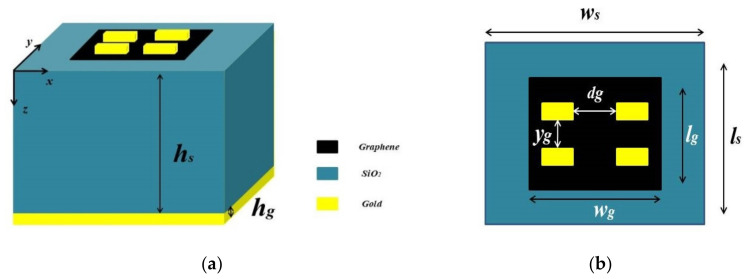
(**a**) 3D view of the graphene based absorber. (**b**) 2D top view of the presented absorber.

**Figure 2 nanomaterials-12-02693-f002:**
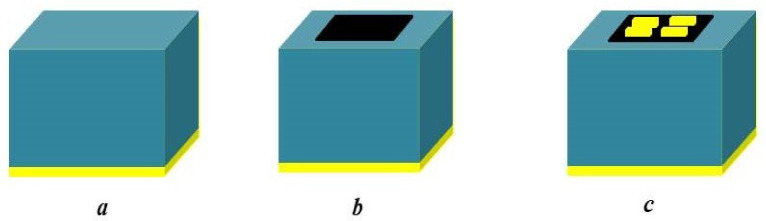
Development stages of the proposed absorber. (**a**) Stage I, (**b**) Stage II (**c**) Final design.

**Figure 3 nanomaterials-12-02693-f003:**
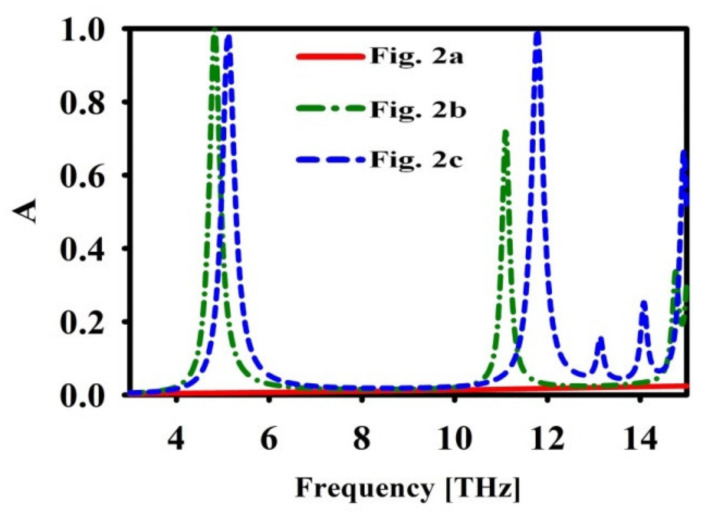
Absorption coefficient (A) of the different structures reported in Figure 2.

**Figure 4 nanomaterials-12-02693-f004:**
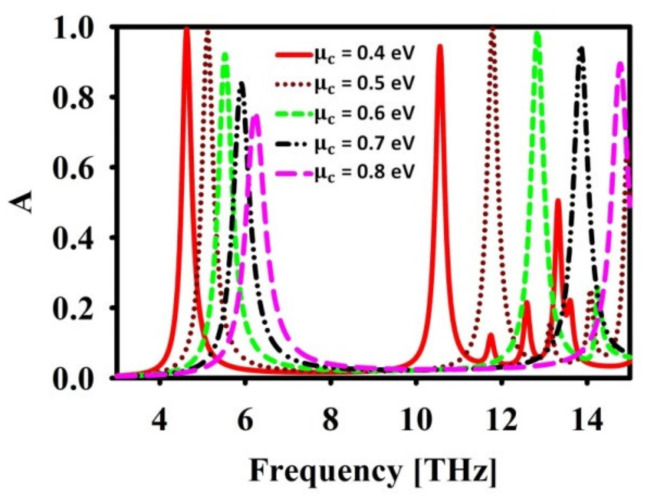
Effect of $\mu_{c}$ in absorption (A) spectra.

**Figure 5 nanomaterials-12-02693-f005:**
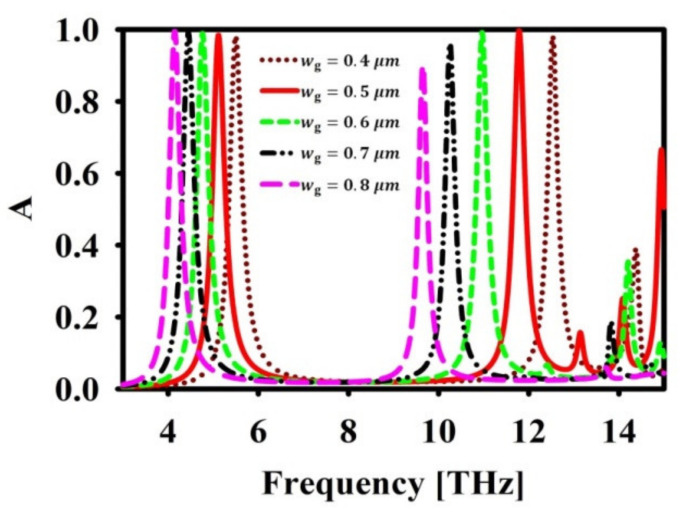
Effect of $w_{g}$ in absorption (A) spectra.

**Figure 6 nanomaterials-12-02693-f006:**
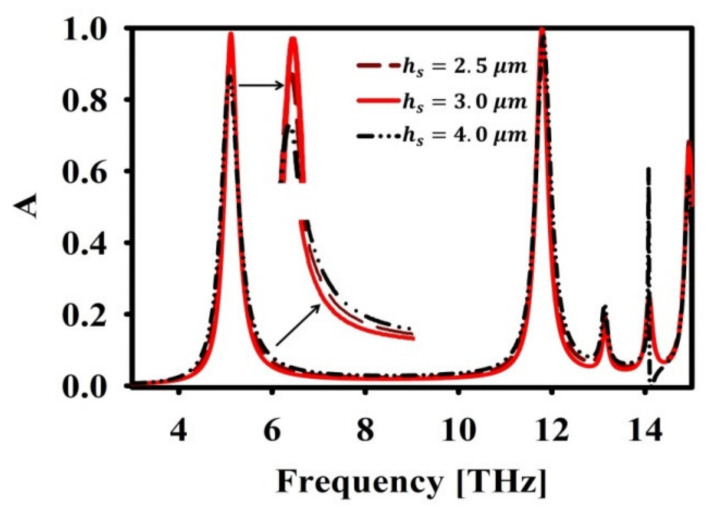
Effect of $h_{s}$ in absorption (A) spectra.

**Figure 7 nanomaterials-12-02693-f007:**
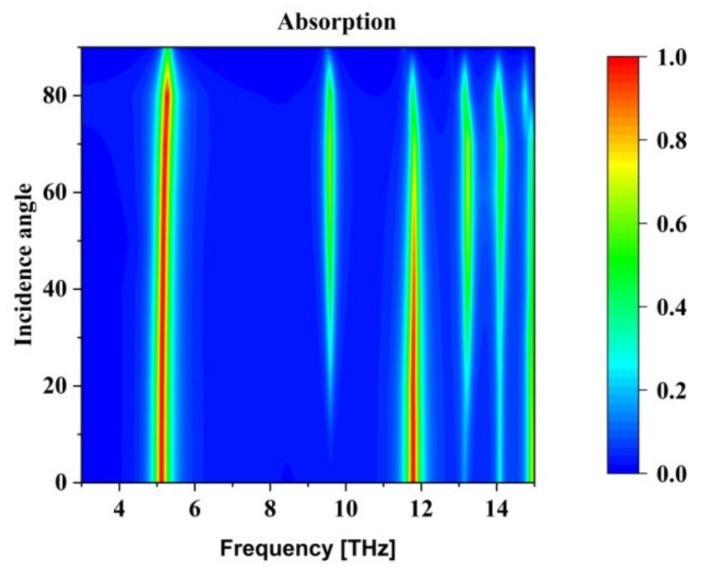
Absorption plot for various incident angles and different frequency.

**Figure 8 nanomaterials-12-02693-f008:**
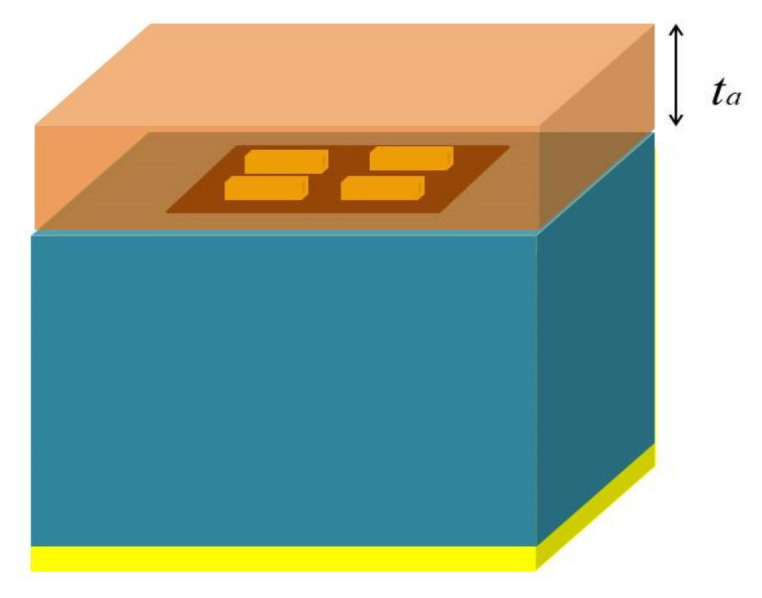
Proposed sensors with analyte thickness of $t_{a}$.

**Figure 9 nanomaterials-12-02693-f009:**
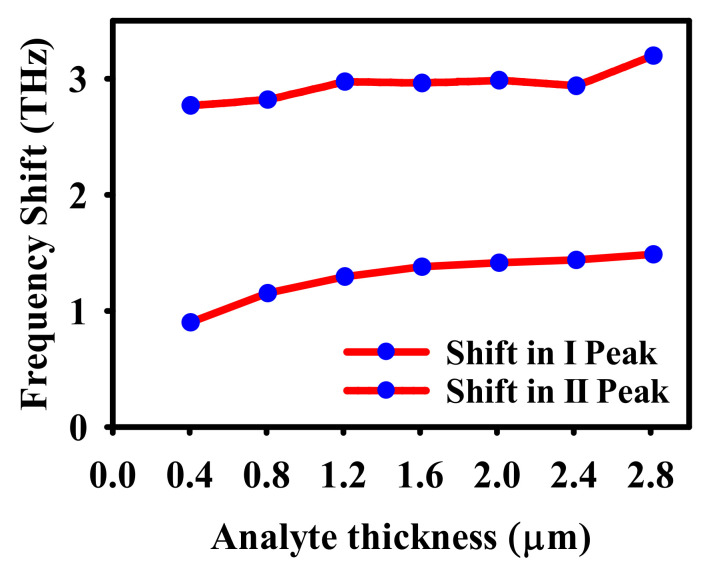
Dependency of frequency shift on analyte thickness.

**Figure 10 nanomaterials-12-02693-f010:**
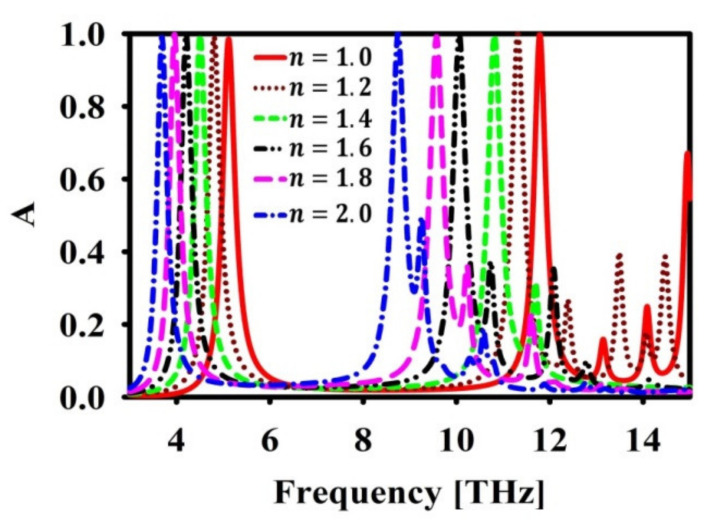
Effect of test medium refractive index $\left(n\right)$ in absorption (A) spectra.

**Figure 11 nanomaterials-12-02693-f011:**
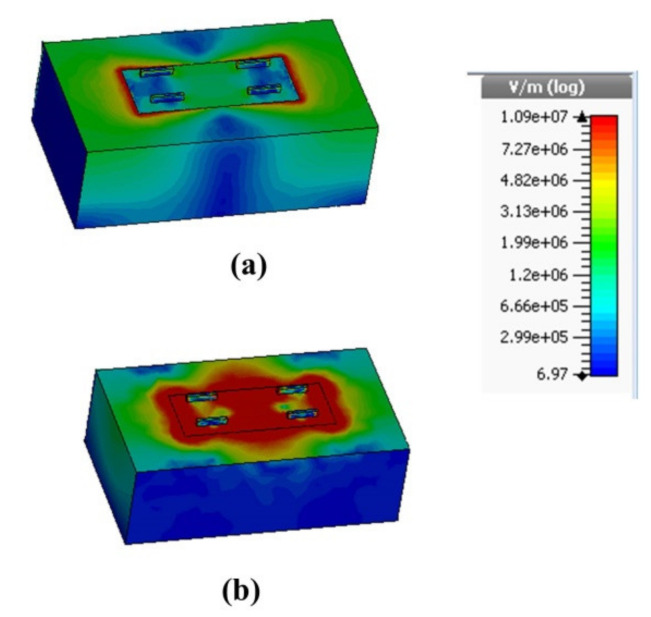
(**a**) Electric field magnitude distribution monitored at I Peak. i.e., f=5.1 THz and (**b**) Electric field distribution monitored at II Peak. i.e., f=11.7 THz.

**Table 1 nanomaterials-12-02693-t001:** Proposed absorber’s optimized dimensions.

Parameter	Value	Parameter	Value
$w_{s}$	9 μm	$d_{g}$	2 μm
$l_{s}$	5 μm	$y_{g}$	0.9 μm
$w_{g}$	5 μm	$h_{s}$	3 μm
$l_{g}$	2 μm	$h_{g}$	0.2 μm
$\mu_{c}$	0.5 ev	T	300 K
τ	1 ps	$t_{g}$	0.34 nm

**Table 2 nanomaterials-12-02693-t002:** Performance of the proposed absorber for sensing 25% glucose in water.

		Δf $\left(\mathrm{THz}\right)$	FWHM $\left(\mathrm{THz}\right)$	Δn	S $\left(\mathrm{THz}/\mathrm{RIU}\right)$	*FOM*	Q
Refractive index changes from *n_w_* = 1.3198 to *n_g_* = 1.3594	I Resonant Peak	0.0827	0.31	0.396	2.08	6.70	14
	II Resonant Peak	0.187	0.34	0.396	4.72	13.88	32.49

**Table 3 nanomaterials-12-02693-t003:** Proposed sensor’s sensing parameters for malaria diagnosis.

n Changes from 1.399 to	S $\left(\mathrm{THz}/\mathrm{RIU}\right)$		FOM $\left({\mathrm{RIU}}^{-1}\right)$		Q	
	I Peak	II Peak	I Peak	II Peak	I Peak	II Peak
1.383	1.52	3.42	4.90	10.05	15.36	34.13
1.373	1.76	3.72	5.67	10.94	15.43	34.28

**Table 4 nanomaterials-12-02693-t004:** Performance evaluation of the suggested absorber in contrast to the existing absorbers.

Ref.	No of Bands	S (I Peak, II Peak) $\left(\mathrm{THz}/\mathrm{RIU}\right)$	*FOM* $\left({\mathrm{RIU}}^{-1}\right)$	f $\left(\mathrm{THz}\right)$	Tunablility	Polarization Insensitive upto the Angle in °	Thickness in μm
[27]	1	0.16	62.82	3.82	Yes	Not reported	30
[35]	1	1.9	6.56	11.5	Yes	Not reported	1
[45]	1	0.13	1.04	3.7	No	Not reported	5
[46]	1	1.48	24.6	3	No	Not reported	10
[25]	1	4.7	13.84	5.71	Yes	70	3
[43]	2	0.22, 3.5	1.1, 7	0.71, 6.4	No	Insensitive	2
[47]	2	0.0001, –	–	0.5, 1.425	No	Not Reported	642
[48]	2	0.1875, 0.360	7.2, 19.1	1.8, 2.26	No	30	8.6
[49]	2	0.147, 0.964	1.21, 17.28	1.08, 2.77	No	Not Reported	10.4
[50]	2	1.35, 2.95	4.67, 8.8	4.88, 10.9	Yes	Insensitive	2.52
This work	2	2.08 4.72	6.7, 13.88	5.1, 11.7	Yes	80	3

## Data Availability

Not applicable.

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
