# Peer review of "Tunable Optimal Dual Band Metamaterial Absorber for High Sensitivity THz Refractive Index Sensing"

_nanomaterials, 2022, doi:10.3390/nano12152693_

Round 1

Reviewer 1 Report

Comments on the manuscript

In this manuscript entitled “Tunable optimal dual band metamaterial absorber for high sensitivity THz refractive index sensing” the authors presented theoretical simulation results of perfect absorbers (dual-band) based on the metal-insulator-graphene-metal configuration. They included detailed studies on the influence of many geometric parameters on the performance of their devices, such as graphene chemical potential, the width of the graphene sheet, the thickness of silicon dioxide spacer, incident angles, and many others. The manuscript, in general, is interesting and their simulation results are incremental to the field of graphene perfect absorbers and graphene sensors. Besides, the manuscript is technically sound with well-supported conclusions and assertions. However, the language is a little difficult to understand, with flaws in both fluency and accuracy. In addition, the authors only presented superficial simulation results and the manuscript lacks physical insight, which degrades the whole level of the manuscript. Thus, the key argument points of the manuscript are weak.

Thus, this manuscript needs substantial revision before it can meet the scope of Nanomaterials. My comments and suggestion to the authors are shown below.

1.       My major concern is the novelty of the manuscript. The idea of “perfect absorbers + graphene +sensors” is not new. Similar contents have been investigated extensively. For example ["Dual-band tunable perfect absorber based on monolayer graphene pattern." Results in Physics 18 (2020): 103306]. Thus what is the key selling point of this work? What are your uniqueness and advantages over previous studies? The authors need to think seriously about these comments and questions, and briefly highlight the selling point in the abstract since it is closely related to the technical innovations and scientific impact of the manuscript.

2.       Language issue. “The most interesting property of graphene is the perfect absorption of terahertz (THz) radiation makes it suitable for applications like sensors [4], detectors[5], cloaking devices [6], thermal emitters [7], modulators[8] and so on.” This sentence is Grammarly incorrect, and it can be revised as “The most interesting property of graphene is the perfect absorption of terahertz (THz) radiation, which makes it suitable for applications like sensors [4], detectors[5], cloaking devices [6], thermal emitters [7], modulators[8] and so on.

3.       “Recently, researchers have been looking into the use of ultrathin [9], ultrasensitive absorption-based sensors [10], [11] and biosensors with a narrow [12]and ultra-narrowband response [13], [14].” Some recent works on ultra-narrow band absorbers are missing. For example ["Hybrid anisotropic plasmonic metasurfaces with multiple resonances of focused light beams." Nano Letters 21.20 (2021): 8917-8923].

4.       “To lessen plasmonic wave power loss, Vanani et al. have introduced a double waveguide THz sensor made of graphene and GaAs. Instead of using the standard double waveguide mode, they have placed a tiny ring in place of one of the waveguides to increase the structure's sensitivity by using constrained supermodes [20]. ” The physics of bound states in the continuum is an essential mechanism for improving the Q-factor in plasmonic structure. The authors may highlight this point in the introduction using one or two sentences if they think it is suitable.

5.       Figure issue. “Fig. 9 Dependency of frequency shift on analyte thickness.” Please show the unit of the thickness (μm or others?).

6.       “Figure 8 depicts the sensing process using analyte medium. The properties of graphene surface plasmons change with changing refractive index of the analyte medium.” The analyte film is an external cavity for the proposed metasurfaces. Have the authors considered the coupling effects between the analyte film and the metasurfaces once the film thickness increase to a higher value? 

Author Response

Point 1: In this manuscript entitled “Tunable optimal dual band metamaterial absorber for high sensitivity THz refractive index sensing” the authors presented theoretical simulation results of perfect absorbers (dual-band) based on the metal-insulator-graphene-metal configuration. They included detailed studies on the influence of many geometric parameters on the performance of their devices, such as graphene chemical potential, the width of the graphene sheet, the thickness of silicon dioxide spacer, incident angles, and many others. The manuscript, in general, is interesting and their simulation results are incremental to the field of graphene perfect absorbers and graphene sensors. Besides,the manuscript is technically sound with well-supported conclusions and assertions. However, the language is a little difficult to understand, with flaws in both fluency and accuracy. In addition, the authors only presented superficial simulation results and the manuscript lacks physical insight, which degrades the whole level of the manuscript. Thus, the key argument points of the manuscript are weak. Thus, this manuscript needs substantial revision before it can meet the scope of Nanomaterials. My comments and suggestion to the authors are shown below.

Response 1: We appreciate your thoughtful advice on how to make our paper better. We have addressed each of your review comments specifically in the text below, as you requested. 

Point 2: My major concern is the novelty of the manuscript. The idea of “perfect absorbers + graphene +sensors” is not new. Similar contents have been investigated extensively. For example ["Dual-band tunable perfect absorber based on monolayer graphene pattern." Results in Physics 18 (2020): 103306]. Thus what is the key selling point of this work? What are your uniqueness and advantages over previous studies? The authors need to think seriously about these comments and questions, and briefly highlight the selling point in the abstract since it is closely related to the technical innovations and scientific impact of the manuscript.

Response 2: The suggested work's relatively straightforward structure and small size are its key advantages. Additionally, the thickness of the 5  reference paper you provided (["Dual-band tunable perfect absorber based on monolayer graphene pattern." Results in Physics 18 (2020): 103306].) has a maximum sensitivity of 1.04 THz/RIU at the lower peak and 2.34 THz/RIU at the upper peak, respectively. The proposed design's maximum sensitivity at the lower and higher peaks, respectively, is 2.08 THz/RIU and 4.72 THz/RIU, but the tightness is only 3 . Additionally, we tested our suggested design in two various sensing contexts. All the mentioned advantages are included in the abstract.

Point 3:  Language issue. “The most interesting property of graphene is the perfect absorption of terahertz (THz) radiation makes it suitable for applications like sensors [4], detectors[5], cloaking devices [6], thermal emitters [7], modulators[8] and so on.” This sentence is Grammarly incorrect, and it can be revised as “The most interesting property of graphene is the perfect absorption of terahertz (THz) radiation, which makes it suitable for applications like sensors [4], detectors[5], cloaking devices [6], thermal emitters [7], modulators[8] and so on.”

Response 3: Thanks for mentioning this mistake. It is corrected in the main document and the entire document is checked for grammatical and typing mistakes.

Point 4:  “Recently, researchers have been looking into the use of ultrathin [9], ultrasensitive absorption-based sensors [10], [11] and biosensors with a narrow [12]and ultra-narrowband response [13], [14].” Some recent works on ultra-narrow band absorbers are missing. For example ["Hybrid anisotropic plasmonic metasurfaces with multiple resonances of focused light beams." Nano Letters 21.20 (2021): 8917-8923].

Response 4: Thanks for mentioning this reference paper. It has been included in the literature survey as well as in the reference list

Point 5:  “To lessen plasmonic wave power loss, Vanani et al. have introduced a double waveguide THz sensor made of graphene and GaAs. Instead of using the standard double waveguide mode, they have placed a tiny ring in place of one of the waveguides to increase the structure's sensitivity by using constrained supermodes [20]. ” The physics of bound states in the continuum is an essential mechanism for improving the Q-factor in plasmonic structure. The authors may highlight this point in the introduction using one or two sentences if they think it is suitable.

Response 4: We appreciate you bringing this to our attention. The following lines have been added in introduction per your recommendation.

“One of the strong interactions that has recently been included into the metasurface system is bound states in the continuum (BIC), which is capable of encapsulating electromagnetic wave energy in the metasurface without radiation leakage [26]”

Point 5:   Figure issue. “Fig. 9 Dependency of frequency shift on analyte thickness.” Please show the unit of the thickness (μm or others?).

 Response 5: Thanks for highlighting this mistake. The unit of thickness is added in the figure’s x-axis as μm .

Point 6:    “Figure 8 depicts the sensing process using analyte medium. The properties of graphene surface plasmons change with changing refractive index of the analyte medium.” The analyte film is an external cavity for the proposed metasurfaces. Have the authors considered the coupling effects between the analyte film and the metasurfaces once the film thickness increase to a higher value? 

      Response 6: As can be seen from figure 9, the frequency shift becomes saturated at a thickness of 2 μm , therefore we assessed the impact of sensing dependence up to this point. Hence, we did not evaluate the impacts of a larger analyte thickness value.

We appreciate your thoughtful remarks. Without a doubt, this will improve the quality of our manuscript. We look forward to the reviewer's suggestions if there is room for improvement.

Reviewer 2 Report

In this paper, the authors report a simple dual band absorber and study its performance in terahertz (THz) region. The proposed absorber is working in dual operating bands at 5.1 THz and 11.7 THz. The proposed sensor is simulated for glucose detection and observed the maximum sensitivity of 4.72 THz/RIU. It has a maximum figure of merit (FOM) and Quality factor (Q) value of 14 and 32.49, respectively. The proposed optimal absorber can be used to identify malaria virus and cancer cells in blood. The proposed plasmonic sensor is a serious contender for biomedical uses in the diagnosis of bacterial infections, cancer, malaria, and other diseases. I believe that publication of the manuscript may be considered only after the following issues have been resolved.

1.      In the model designed by the author, the size of graphene has area requirements. Can graphene be designed to be infinite?

2.      The big problem of the article is that many pictures are simple descriptions, and there is no in-depth analysis of why they produce such changes. It is suggested that the author should make detailed supplements.

3.      The font of the color column in Figure 11 is too small, and the author needs to adjust it.

4.      When table 2 and table 3 are used for detection and analysis of different substances, it is suggested that the author still use curves to reflect them.

5.      In the part of design structure and model, some structures and details are not well described by the author. Such as, what does tg stand for? What are the conductivity parameters of Au?

6.      The introduction can be improved. The articles related to the metamaterial absorbers and their related applications should be added such as Phys. Chem. Chem. Phys., 2022, 24, 8846 8853; Plasmonics 2015, 10, 15371543; RSC Adv., 2022, 12(13), 7821-7829; Plasmonics 2018, 13, 345352. Appl. Phys. Express 2019, 12, 052015.

Author Response

Point 1: In this paper, the authors report a simple dual band absorber and study its performance in terahertz (THz) region. The proposed absorber is working in dual operating bands at 5.1 THz and 11.7 THz. The proposed sensor is simulated for glucose detection and observed the maximum sensitivity of 4.72 THz/RIU. It has a maximum figure of merit (FOM) and Quality factor (Q) value of 14 and 32.49, respectively. The proposed optimal absorber can be used to identify malaria virus and cancer cells in blood. The proposed plasmonic sensor is a serious contender for biomedical uses in the diagnosis of bacterial infections, cancer, malaria, and other diseases. I believe that publication of the manuscript may be considered only after the following issues have been resolved.

Response 1: We value your insightful suggestions on how to improve our paper. As per your suggestion, we have specifically responded to each of your review comments in the paragraph that follows.

Point 2: In the model designed by the author, the size of graphene has area requirements. Can graphene be designed to be infinite?

Response 2: yes it's impossible. Here, infinity is referred in terms of periodic boundary conditions (PBCs), which are frequently used in simulations of molecular dynamics. Additionally, we would like to clarify regarding the graphene fabrication on SiO2 and hence added the following paragraph in the main document.

“This structure can be constructed using a clear and precise method that is fully described in [55]. Using a shadow mask during the graphene's plasma enhanced chemical vapour deposition (PECVD), the suggested method makes it possible to fabricate graphene in any shape on a SiO2 substrate. Other strategies and approaches for producing graphene with dimensions less than a nanometer have been researched into and presented in [56]”

Point 3: The big problem of the article is that many pictures are simple descriptions, and there is no in-depth analysis of why they produce such changes. It is suggested that the author should make detailed supplements.

Response 3: Our goal is to demonstrate the proposed absorber's advantages in terms of numerical value and assert that it is comparatively superior to other designs. The results and comments include a detailed analysis of the proposed absorber's performance.

Point 4: The font of the color column in Figure 11 is too small, and the author needs to adjust it.

     Response 4:  Thanks for highlighting this correction. The font of color column is increased.

Point 5: When table 2 and table 3 are used for detection and analysis of different substances, it is suggested that the author still use curves to reflect them.

Response 5:  Thanks for your suggestions. However, in order to provide a basis for relation to the existing design, we attempt to provide the suggested design's real values for sensitivity, FOM, and Q-Factor.

Point 6: In the part of design structure and model, some structures and details are not well described by the author. Such as, what does tg stand for? What are the conductivity parameters of Au?

Response 6:  We would like to make it clear that tg refers to the graphene thickness and the conductivity of the Au  is , which are both included in the initial submission itself under the heading "Design structure and model."

Point 7: The introduction can be improved. The articles related to the metamaterial absorbers and their related applications should be added such as Phys. Chem. Chem. Phys., 2022, 24, 8846 – 8853; Plasmonics 2015, 10, 1537–1543; RSC Adv., 2022, 12(13), 7821-7829; Plasmonics 2018, 13, 345–352. Appl. Phys. Express 2019, 12, 052015.

Response 7:  Thank you for making us aware of this papers. It was quite informative. The contribution of these works is covered in the introduction section.

We value your thoughtful comments very much. This will unquestionably raise the calibre of our manuscript. If there is room for improvement, we anticipate the reviewer's remarks.

Round 2

Reviewer 1 Report

The authors have addressed my comments and improved the manuscript substantially. The manuscript is scientifically sound. Thus, I have no further questions but to give my proposal of acceptance. Good luck to the authors.

Reviewer 2 Report

Accept in present form.